The European and Japanese outbreaks of H5N8 derive from a single source population providing evidence for the dispersal along the long distance bird migratory flyways

Dalby Andrew R. 1 A.Dalby@westminster.ac.uk
Iqbal Munir 2
1 Faculty of Science and Technology, University of Westminster , Westminster , UK
2 Avian Viral Diseases Programme, Compton Laboratory, Pribright Institute , Newbury, Berkshire , UK
Wilke Claus
Electronic publication date: 2015 Apr 30
Publication date: 2015
Volume: 3
Electronic Location ID: e934
Received 2015 Feb 8; Accepted 2015 Apr 14
Copyright: © 2015 Dalby and Iqbal
Copyright year: 2015
Copyright holder: Dalby and Iqbal
License: This is an open access article distributed under the terms of the Creative Commons Attribution License, which permits unrestricted use, distribution, reproduction and adaptation in any medium and for any purpose provided that it is properly attributed. For attribution, the original author(s), title, publication source (PeerJ) and either DOI or URL of the article must be cited.
License URL: https://creativecommons.org/licenses/by/4.0/

Keywords: High pathogenicity avian influenza, HPAI, H5N8, Migratory birds, Siberia, Korea, Bayesian, Coalescence

Funding: The authors declare there was no funding for this work.

==============================
The origin of recent parallel outbreaks of the high pathogenicity H5N8 avian flu virus in Europe and in Japan can be traced to a single source population, which has most likely been spread by migratory birds. By using Bayesian coalescent methods to analyze the DNA sequences of the virus to find the times for divergence and combining this sequence data with bird migration data we can show the most likely locations and migratory pathways involved in the origin of the current outbreak. This population was most likely located in the Siberian summer breeding grounds of long-range migratory birds. These breeding grounds provide a connection between different migratory flyways and explain the current outbreaks in remote locations. By combining genetic methods and epidemiological data we can rapidly identify the sources and the dispersion pathways for novel avian influenza outbreaks.

Introduction

The H5N8 subtype of influenza A is a comparatively rare influenza A subtype that was first isolated from a turkey in Ireland in 1983 (Murphy, 1986). After that initial outbreak there were no more cases of H5N8 until 2001, when a case was identified during environmental monitoring in a wild bird in New Jersey. Since then there have been a few sporadic detections in the United States, but the biggest single outbreak to date has been in Korea in January 2014 (Lee et al., 2014).

This outbreak was preceded by cases in Eastern China in 2010 that are distinct from the American virus. Although the Korean outbreak strains had the same subtype, the Asian outbreak seems to have been the product of a re-assortment of viral segments from other H5 containing subtypes such as H5N1 or H5N5 and an N8 containing subtype, rather than from the evolution of the previous H5N8 lineages (Lee et al., 2014; Zhao et al., 2013). The Korean outbreak has been subdivided into two lineages one of which is closely related to the Chinese sequences and that has only been identified in two samples in Gochang and a second that contains all the other cases and that was originally identified in Buan (Fan et al., 2014; Jeong et al., 2014).

The H5N8 virus is an example of a highly pathogenic avian influenza A (HPAI). These HPAI viruses pose a significant threat to domestic poultry, as mortality rates amongst chickens are particularly high and can reach 100%. Recent studies of the virus have shown that it has a pathogenicity index of 3 in chickens (Kim et al., 2014). This is significantly higher than that of the original H5N8 from Ireland, although that is a distinct lineage (Alexander, Parsons & Manvell, 1986). The management of the outbreak in Korea in early 2014 resulted in the culling of over 10 million birds, or 6% of the total Korean poultry flock (Kang et al., 2015). Ducks and particularly wild ducks such as mallards are often asymptomatic but can still be carriers of the H5N8 virus (Bae et al., 2014; Kang et al., 2015; Kim et al., 2014).

Currently unlike H5N1, H5N8 is not considered a threat to human health, as there has not been a case of transmission to humans. However, this might be a result of the low incidence of the subtype as studies have shown that it can be transmitted to ferrets and mice, and antibodies have been detected in domestic dogs (Kim et al., 2014). The results of genetic analysis of the H5N8 virus in infected ferrets have also shown that mutations to a mammalian transmissible form occur rapidly.

In November 2014, H5N8 was detected in Europe, with outbreaks in poultry farms in the Netherlands, Germany and the United Kingdom. At the same time, the virus was also detected in farmed birds and wild birds in Japan. This study identifies the probable geographical source and pathway for dispersal of the November/December outbreaks of H5N8.

Materials and Methods

The complete set of available H5N8 nucleotide sequences were downloaded from the NCBI Influenza Virus Resource and GISAID (Bao et al., 2008; Bogner et al., 2006). The search was restricted to complete sequences of H5N8 within the NCBI influenza virus resource.

All of the sequences were aligned with Muscle v3.8.31 (Edgar, 2004). Manual inspection and editing of the sequences was carried out using Mega6.06 (Tamura et al., 2013). During manual editing, the 5′ end of the sequence was edited to remove the un-translated region. All sequences begin at the start codon. Sequences with missing nucleotides were removed; this included 3 partial Chinese duck sequences. There was no editing at the 3′ end of the nucleotide sequences, as influenza uses a variety of stop codons that are sometimes repeated. Tip dates were assigned according to the year of collection.

A subset of the sequences was created for the detailed analysis of the hemagglutinin and neuraminidase sequences containing only the sequences from 2014; this provides a more detailed analysis for calculating the divergence dates from the Korean outbreak. For these calculations, tip dates were given in months before December 2014. This solves the problem of missing data from earlier sequences where months might not be available.

Bayesian Coalescent trees were calculated for all the different segments and the subsets using Beast2.1.3 (Bouckaert et al., 2014). The model used assumed an exponential population growth, and tip dates were set from the sequence collection dates. The Hasegawa–Kishino–Yano (HKY) nucleotide substitution model was used with an assumed strict molecular clock (Hasegawa, Kishino & Yano, 1985) as using a strict clock was shown to give the best effective sample size when compared to other clock models, which gave similar parameter values but also showed significant auto-correlation. The use of the HKY model in preference to Tamura-Nei in preference to Tamura-Nei was also supported by analysis using Model-Test (Posada & Crandall, 1998). All the final simulations were performed as a single run with a minimum of 10 million iterations, except for the PA segment that needed 20 million, the PB2 segment that needed 30 million and the NS segment that needed 40 million iterations to achieve a suitable level of chain sampling. A burn in period of 10% was used for all samples.

Analysis of the Bayesian coalescent output was carried out using Tracer1.6.0 (Rambaut & Drummond, 2013a). All simulations were run until the effective sample sizes for all of the parameters in the model were over 200. The maximum clade credibility trees were calculated using Treeannotater 2.1.2 along with the median node heights, and the final tree diagrams were generated using FigTree1.4.2 (Rambaut, 2007; Rambaut & Drummond, 2013b).

All of the available H5 hemagglutinin subunits and N8 neuraminidase subunits were downloaded from the NCBI Influenza Virus Resource and GISAID (Bao et al., 2008; Bogner et al., 2006). This data was used to carry out a complete phylogenetic analysis for each of the segments. The sequences were initially aligned using MAFFT (Katoh & Standley, 2013). FastTree2.1 was used to create an approximate maximum likelihood tree for all of the sequences. The resulting trees were visualised and annotated with FigTree 1.4.2 (Rambaut & Drummond, 2013a).

All of the XML files used to calculate the trees for each of the segments and for the 2014 hemagglutinin and neuraminidase trees are available as Supplemental Information 9, as are the tree files they produce. The tree files for the complete H5 and N8 segment analysis are also available in Supplemental Information 10 and Supplemental Information 11.

The locations of the H5N8 cases were taken from the EMPRES Global Animal Disease Information System (EMPRES-i), and information about the original reports were sourced from the Avian Flu Diary Blog (http://afludiary.blogspot.co.uk/). The map was created using Google maps and is available from: https://www.google.com/maps/d/edit?mid=zcvUWKLLjKsE.kvYJ1NxAer8k.

Results and Discussion

The Bayesian coalescent analysis of the complete set of H5N8 sequences produces a consistent gene tree, where the same clade structure is produced for all eight segments (Figs. S1 to S8). The current Korean outbreak is the product of a recent viral re-assortment, and so complete trees of the H5 subunits from all of the viral subtypes and of the N8 subunits from all of the viral subtypes were created to confirm that there had not been another re-assortment between the European and Japanese sequences. These results show that the European and Japanese sequences all form a single cluster closely related to, but distinct from those found in the Korean outbreak. This clustering suggests that the viruses are likely to come from a single source population.

Bayesian coalescent analysis of the 2014 sequences also permits the sequence divergence time to be calculated with greater accuracy (Lemey et al., 2009). The Bayesian coalescent trees for the 2014 hemaggluinin gene segments and the neuraminidase gene segments are shown in Figs. 1 and 2, respectively. The bars above the branch points represent the 95% highest posterior density for the age of that node. The x-axis represents the date in months before November 2014. For the viral hemagglutinin gene segment, the cluster of sequences responsible for the current European and Japanese outbreaks diverged between a median value of 2.65 and 6.57 months before November 2014 (95% highest posterior density interval). Only the hemagglutinin and neuraminidase segments are available from the infected German turkey and the median divergence time calculated from the neuraminidase tree is from 1.39 to 6.21 months before November 2014 (95% highest posterior density interval). This is in good agreement with the results from the hemagglutinin tree, and shows that the Japanese and European sequences diverged after the last reported cases of the Korean outbreak.

Figure 1 Bayesian coalescent gene tree for the 2014 H5N8 hemagglutinin sequences.

The blue bars on the nodes represent the 95% highest posterior density intervals of the branch ages. The European and Japanese clade is highlighted in red. 0 on the x-axis represents November 2014.

Figure 2 Bayesian coalescent gene tree for the 2014 H5N8 neuraminidase sequences.

The blue bars on the nodes represent the 95% highest posterior density intervals of the branch ages. The European and Japanese clade is highlighted in red. 0 on the x-axis represents November 2014.

Figure 3 Bird migratory flyways and the December 2014 cases of H5N8 (Boere & Stroud, 2006).

The Eastern Asian Australian flyway is in red. The East Atlantic flyway is in dark blue. An expandable version of this map is available from: https://www.google.com/maps/d/viewer?mid=zcvUWKLLjKsE.kvYJ1NxAer8k—Map Data ©2015 Google, INEGI

During the Korean outbreak, a large number of wild birds were also affected, particularly in the region around the Dong-Lim reservoir (Jeong et al., 2014). One of the bird species that was found to be infected was the Baikal Teal (Anas formosa), which is a migratory species that over winters in Korea before returning to North Eastern Siberia to breed during the summer months (Allport et al., 1991). This migration coincided with the last Korean H5N8 sequences identified in wild birds during the initial outbreak. This migration also falls within the range of divergence dates from the Bayesian coalescent analysis for the current cluster of H5N8 cases in Europe. This result strongly suggests that the virus was carried to the Siberian breeding grounds as the Baikal teal migrated north and that the European and Japanese sequences evolved there.

The wide geographic dispersal of the current outbreaks gives further support to the contention that migratory birds are the source of the virus. Most of the recent cases occur close to the coastline and in areas where there are lakes and known sites for waterfowl and migratory birds. In Holland the virus has been identified in widgeon, and in Germany it was found in a common teal (Anas crecca) that had no apparent clinical symptoms. The Japanese have recently identified the wild bird species infected with the virus as tundra swans (Cygnus columbianus), white naped cranes (Grus vipio), pochards (Aythya ferina) and wild ducks (Anas platyrhyncus).

The breeding grounds and migratory staging grounds for Baikal teal overlap with those for many other migratory species including common species such as mallards, pochards, widgeon (Anas penelope), common teal, whooper swans (Cygnus cygnus) and tundra swans, as well as endangered species such as white-naped cranes (Miyabayashi & Mundkur, 1999). Mallard and teal have previously been identified as having a high prevalence (between 6 and 7%) for influenza A virus (Munster et al., 2007). The bird migrations flow from Siberia along the five different flyways that overlap in Central Siberia; they are the East Atlantic, East Asia Australian, East Africa West Asia, Central Asia and Black Sea Mediterranean flyways. So far, H5N8 infections in birds have only been detected in the East Atlantic and East Asian Australian flyways (Fig. 3). The absence from other flyways can be explained either through transmission by a limited number of bird migratory species, or because or the lack of surveillance in these geographical regions. Although there have been a few cases recently reported in North America, these are from the different lineage not related to the Korean outbreak.

This year, the winter migration has been later than usual because of the warmer autumn weather. Ideally, satellite-tracking data would be available for all of the migrating species from their summer breeding grounds. However, tracking data is only available for species of interest that include Bewick swans (Cygnus bewickii), a sub-species of Tundra swans. These tracking data show that their migration was delayed until late October and early November, which coincided with the European outbreaks of the H5N8 virus (Slimbridge Wildlife Trust).

Gaidet and co-workers (2010) considering the spread of another HPAI H5N1, suggested that the risk of transmission by migratory birds was only a low risk because of the need for asymptomatic infections and also taking into account the distances travelled, the time taken and the number of staging points along the journey. However, low pathogenicity avian influenza have been shown to spread via migrating birds because the large majority of cases remain asymptomatic (Dusek et al., 2014; Lam et al., 2012). In the case of high pathogenicity H5N8, the virus has been shown to be asymptomatic in mallards, and there would have been selection of virus variants that are asymptomatic amongst the Baikal teal if the disease has been carried by a migrant bird (Bae et al., 2014; Kang et al., 2015; Kim et al., 2014). Dispersion of the virus through migratory flyways still requires that there is relay infection for the virus to spread over very long migratory distances.

Previous studies had shown that there was a spatio-temporal relationship between bird migration and the spread of the HPAI H5N1 subtype (Takekawa et al., 2010). However, it was not possible to show that transmission by the migratory birds was the cause of this correlation. In this case, the genetic data and the calculated divergence times show that the evolutionary events responsible for generating the European and Japanese cases occurred in the summer months in a single location.

The initial outbreak affected a large number of birds during the period close to the main spring migration; this increased the likelihood of long-range transmission. The spread of the virus requires that there is relay infection so that it spreads amongst susceptible birds at the migratory staging points, in order to provide the next step in transmission. This is seen with the presence of an increased number of cases at staging points such as the Netherlands. This conclusion is supported by the current limited amount of data, although there are other staging points in Estonia and in Denmark where there have not been any reported cases (Beekman, Nolet & Klaassen, 2002; Green et al., 2002).

Conclusions

The results presented here give strong support to the view that the H5N8 outbreaks that occurred in Europe and Japan in December 2014 originated from a single source population. Although there is no direct evidence of what this source population was, it is likely that the virus was spread along long-range migratory routes as trade, is a less likely source given the absence in disease infections during the summer months. This suggests that the summer breeding grounds for migratory species such as Baikal teal are the most likely geographical location for the source of the outbreaks.

Increased monitoring for HPAI is needed in areas where there is overlap between migrating species, especially if this zone links very disparate geographical regions. This could be achieved through environmental monitoring of faecal samples in areas where migratory birds congregate. In the case of H5N8, the main costs are economic as it is not currently a human pathogenic subtype, but it has had devastating consequences for the Korean poultry industry. However, the longer the virus is in circulation in wild birds and poultry, the more likely it is that a human case will occur, especially considering the close relationship to the H5N1 strains and existing evidence that shows the virus can reproduce in mammalian hosts.

It is important to involve local communities and experts as well as farmers so that we can significantly improve the monitoring network giving earlier warning of potential epidemics. This also means improved communication between international organizations and making the biological sequence data available in a timely manner.

Supplemental Information

Figure S1 Bayesian coalescent tree for hemagglutinin

Bayesian coalescent gene tree for all of the H5N8 hemagglutinin segment sequences. The blue bars on the nodes represent the 95% highest posterior densities of the branch heights (this is the time for divergence in months). The European and Japanese clade is highlighted in red. The x-axis represents the date in years.

Click here for additional data file.

Figure S2 Bayesian coalescent tree of neuraminidase

Bayesian coalescent gene tree for all of the H5N8 neuraminidase segment sequences. The blue bars on the nodes represent the 95% highest posterior densities of the branch heights (this is the time for divergence in months). The European and Japanese clade is highlighted in red. The x-axis represents the date in years.

Click here for additional data file.

Figure S3 Bayesian coalescent tree of the matrix protein segment

Bayesian coalescent gene tree for all of the H5N8 matrix protein segment sequences. The blue bars on the nodes represent the 95% highest posterior densities of the branch heights (this is the time for divergence in months). The European and Japanese clade is highlighted in red. The x-axis represents the date in years.

Click here for additional data file.

Figure S4 Bayesian coalescent tree of polymerase protein segment PB1

Bayesian coalescent gene tree for all of the H5N8 polymerase subunit (PB1) segment sequences. The blue bars on the nodes represent the 95% highest posterior densities of the branch heights (this is the time for divergence in months). The European and Japanese clade is highlighted in red. The x-axis represents the date in years.

Click here for additional data file.

Figure S5 Bayesian coalescent tree for the polymerase protein segment PB2

Bayesian coalescent gene tree for all of the H5N8 polymerase subunit (PB2) segment sequences. The blue bars on the nodes represent the 95% highest posterior densities of the branch heights (this is the time for divergence in months). The European and Japanese clade is highlighted in red. The x-axis represents the date in years.

Click here for additional data file.

Figure S6 Bayesian coalescent tree for the polymerase PA protein segment

Bayesian coalescent gene tree for all of the H5N8 polymerase subunit (PA) segment sequences. The blue bars on the nodes represent the 95% highest posterior densities of the branch heights (this is the time for divergence in months). The European and Japanese clade is highlighted in red. The x-axis represents the date in years.

Click here for additional data file.

Figure S7 Bayesian coalescent tree for the nucleoprotein segment

Bayesian coalescent gene tree for all of the H5N8 nucleoprotein subunit (NP) segment sequences. The blue bars on the nodes represent the 95% highest posterior densities of the branch heights (this is the time for divergence in months). The European and Japanese clade is highlighted in red. The x-axis represents the date in years.

Click here for additional data file.

Figure S8 Bayesian coalescent tree for the non-structural protein segment

Bayesian coalescent gene tree for all of the H5N8 non-structural protein subunit (NS) segment sequences. The blue bars on the nodes represent the 95% highest posterior densities of the branch heights (this is the time for divergence in months). The European and Japanese clade is highlighted in red. The x-axis represents the date in years.

Click here for additional data file.

Supplemental Information 9 zip file of all the xml files the complete H5 and N8 tree files

Click here for additional data file.

Supplemental Information 10 Hemagglutinin trees for H5N8

Click here for additional data file.

Supplemental Information 11 Neuraminidase trees for H5N8

Click here for additional data file.

Supplemental Information 12 Segment polymerase A trees

Click here for additional data file.

Supplemental Information 13 Segment PB1 trees

Click here for additional data file.

Supplemental Information 14 Segment PB2 trees

Click here for additional data file.

Supplemental Information 15 Segment MP trees

Click here for additional data file.

Supplemental Information 16 Segment NP trees

Click here for additional data file.

Supplemental Information 17 Segment NS trees

Click here for additional data file.

Special thanks must go to Michael Coston, whose Avian Flu Diary Blog has been an important source of information for the current outbreak. Also thanks to David Welch for his comments on the Bayesian analysis. Finally, thanks to the editor Prof. Claus Wilke and the anonymous referee who pointed us towards a previously unexplored area for our research.

Additional Information and Declarations

Competing Interests

Author Contributions

Data Deposition

The authors declare there are no competing interests.

Andrew R. Dalby conceived and designed the experiments, performed the experiments, analyzed the data, contributed reagents/materials/analysis tools, wrote the paper, prepared figures and/or tables, reviewed drafts of the paper.

Munir Iqbal wrote the paper, reviewed drafts of the paper.

The following information was supplied regarding the deposition of related data:

Google Maps: https://www.google.com/maps/d/viewer?mid=zcvUWKLLjKsE.kvYJ1NxAer8k.

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
