# Peer review of "The European and Japanese outbreaks of H5N8 derive from a single source population providing evidence for the dispersal along the long distance bird migratory flyways"

_PeerJ, doi:10.7717/peerj.934_

## Round 0.1 · original submission · Major Revisions

Both reviewers raise good points that should be addressed.

·

Basic reporting

No comment

Experimental design

Beast analyses are easily reproduced by making the XML files available. These should be added as supplementary materials.

Need some detail about length of sequences.

SOme of the methodology described does not meet the highest standards of phylogenetic analysis (for example, there is no need to remove sequnces with missing nucleotides (line 55), the population growth model seems arbitrarily chosen and not tested against others (lne 66) and the HKY substitution model is chosen based on it giving high posterior probabilities (should be densities, line 70) which is not a valid criteria). However, these weaknesses are unlikely to change the the main result which is that the European and Japanese sequence form a recent clade.

Validity of the findings

There are two findings here: one is that the 2014 Europoean and Japanese sequences cluster together on the tree. This is solid and easy to show.

The second is that the mechanism of spread from Asia to Europe was most likely a migratory bird. The evidence provided here is largely circumstantial and, although convincing, it is of a different nature to the other result. I would make it clear throughout the distinction in the type of evidence you are presenting here --- it is possible to model geographic spread of sequences directly within the Beast framework and, on reading the title and abstract, that was what I thought you had done. It is not necessary to run a full phylogeoographic model but the tile could be changed to something like "The European and Japanese outbreaks of H5N8 derive from a single source population providing evidence for dispersal along the long distance bird migratory flyways" and similar changes made throughout.

Additional comments

The dates used for the sample times should be consistent throughout. Lines 96 and 98 are inconsistent in this respect. Why not use real dates throughout --- it is asy for you to convert your internal dating method to real dates for presentation. This applies also to figures 1 and 2 --- use real dates on the x-axis.

Lines 101-102 Beast deals with missing nucleotide data --- you don't need to omit sequences with missing data or impute it yourselves (imputing yourselves is a bad idea).

Sentence lines 94-95: the bars represent the 95% highest posterior density (HPD) interval for the age of that node.

Check use of "highest posterior density" throughout --- normal usage is x% HPD interval, or x% credible interval (HPD).

Figures 1 and 2 (and supplemantary figs) should have tips in similar order to make them easier to compare. Also, simplify the tip names so that the tree is the dominant feature rather than the tip labels.

Reviewer 2 ·

Basic reporting

No comments

Experimental design

1) By selecting only full H5N8 nucleotide sequences, the author’s will be ignoring the possibility that there are other H5 subtypes that are closely related to the Japanese and European lineage. Likewise for N8 subtypes. It would be advisable to download all available H5 full-length sequences and produce a fast maximum-likelihood tree to determine more comprehensively what the closest lineage to the Japanese and European H5N8 is. Likewise for all N8 subtypes.
2) Considering the phylogeographic focus of the manuscript, the authors should justify why they have not used a standard phylogeographgic analysis in BEAST.

Validity of the findings

The authors need to show that the data they have used for this study is sufficient to make their conclusions (see experimental design).

Additional comments

This manuscript by Andrew Dalby and Munir Iqbal investigates the origin of the European and Japanese H5N8 outbreaks. It is an interesting area of research and a well-written article. However, due to the selection of sequences for their study, they may be missing important evolutionary context.

Line 40: that that => that
Line 120: where there lakes => where there are lakes

---

## Round 0.2 · Minor Revisions

Thank you for addressing the previous reviewer comments. Both reviewers were mostly satisfied with your revisions. However, Reviewer 1 had a few additional minor comments. Please address the wording issues pointed out by the reviewer, and consider whether you would like to run the additional analyses suggested. I consider the additional analyses optional.

·

Basic reporting

"No Comments"

Experimental design

"No Comments"

Validity of the findings

"No Comments"

Additional comments

Overall the manuscript is much improved and I support publication.

A few details:

Lines 67-70: You say HKY was chosen based on it giving the best ESS. This is not a good reason to choose the model, it just means it runs quicker. You could get the same ESS from other models by running them longer --- run times for this data would not be prohibitive.

Line 68: should be "strict" not "rigid" molecular clock. Anyway, I'd try a relaxed clock here but as I mentioned in previous review, results look robust to this type of model choice.

In several places you use "coalescence tree/analysis". This is usually "coalescent tree/analysis".

Reviewer 2 ·

Basic reporting

pass

Experimental design

pass

Validity of the findings

pass

Additional comments

I am satisfied that the authors have now done the analyses necessary to support their statement that “the European and Japanese sequences all form a single cluster closely related to, but distinct from those found in the Korean outbreak”. This statement can only be made in the context of the full H5 and full N8 phylogenies, which the authors have now conducted.

---

## Round 0.3 · accepted · Accept

I am accepting this manuscript. Note, however, that in the revisions you introduced the word "rigid molecular clock" once more. Please revise this to "strict molecular clock" in the proof.